# Hybrid Ladder Transformers with Efficient Parallel-Cross Attention for Medical Image Segmentation

**Luo Haozhe**[1]                                                                    LUOHAOZHE@STU.SCU.EDU.CN
[1] *Department of Computer Science & Technology, Sichuan University, China*

**Yu Changdong**[2]                                                                    YCD0212@163.COM
[2] *College of Information & Communication Engineering, Harbin Engineering University, China*

**Raghavendra Selvan**[3,4]                                                             RAGHAV@DI.KU.DK
[3] *Department of Computer Science, University of Copenhagen, Denmark*
[4] *Department of Neuroscience, University of Copenhagen, Denmark*

**Editors:** Under Review for MIDL 2022

## Abstract

Most existing transformer-based network architectures for computer vision tasks are large (in number of parameters) and require large-scale datasets for training. However, the relatively small number of data samples in medical imaging compared to the datasets for vision applications makes it difficult to effectively train transformers for medical imaging applications. Further, transformer-based architectures encode long-range dependencies in the data and are able to learn more global representations. This could bridge the gap with convolutional neural networks (CNNs), which primarily operate on features extracted in local image neighbourhoods. In this work, we present a hybrid transformer-based approach for segmentation of medical images that works in conjunction with a CNN. We propose to use learnable global attention heads along with the traditional convolutional segmentation network architecture to encode long-range dependencies. Specifically, in our proposed architecture the local information extracted by the convolution operations and the global information learned by the self-attention mechanisms are fused using bi-directional cross attention during the encoding process, resulting in what we call a *hybrid ladder transformer* (HyLT). We evaluate the proposed network on two different medical image segmentation datasets. The results show that it achieves better results than the relevant CNN- and transformer-based architectures.[1]

**Keywords:** self-attention, transformers, u-net, segmentation

## 1. Introduction

Convolutional neural networks (CNNs) have been the work-horse for computer vision tasks for the past few years. Medical image segmentation with CNN based models such as the U-net (Ronneberger et al., 2015) are some of the most widely used supervised learning methods. These CNN based architectures rely on strong inductive biases and learn local image features using shareable kernels. One of the features of CNNs that is sometimes restrictive is their fixed receptive field (Luo et al., 2016), as CNN kernels operate on small pixel neighbourhoods. This is alleviated to some extent using hierarchical processing (Ronneberger et al., 2015; Zhou et al., 2018; Oktay et al., 2018; Sinha and Dolz, 2020). More recently, there have been attempts to work around this limitation in receptive fields by

---

1. Source code is available at: https://github.com/Roypic/LTUNet

incorporating multiple receptive fields within a single network architecture, such as in Su et al. (2021); Xu et al. (2022).

The success of transformer-based methods in natural language processing (NLP) (Vaswani et al., 2017; Devlin et al., 2019) in the past couple of years has prompted a flurry of research in other domains. Bulk of this research has been on adapting transformers, which primarily operate on sequence-type data, to other modalities. The vision-transformer (ViT) introduced in Dosovitskiy et al. (2021) attempted this for image classification by obtaining sequence of embeddings from image patches. It has since been argued that transformer-type models can capture more global image features and long-range dependencies, as global attention maps enable them to maintain global representations in all the layers (Dosovitskiy et al., 2021). One of the limitations of ViT-based methods is their dependency on large training datasets and huge compute requirement (Touvron et al., 2021) which make their use for medical image analysis infeasible, where the data are scarce. This has been addressed to some extent with efficient transformer-only architectures such as the Swin-Unet (Cao et al., 2021).

Most recently, there have been several attempts at combining the strengths of CNNs and transformers. Methods such as Detection Transformer (DETR) (Carion et al., 2020) and hybrid Dense Prediction Transformer (DPT) (Ranftl et al., 2021) have attempted to combine local features obtained by CNNs and global information obtained by transformers showing promising results. There have been equally many attempts in the field of medical image segmentation, with works such such as TransUNet (Chen et al., 2021a), Medical Transformer (MEDT) (Valanarasu et al., 2021), UTNet (Gao et al., 2021). The increase in the model complexity (number of parameters) and constraints on fixed input image sizes in these methods are some of the limitations we aim to address in our work.

In this work, we propose an efficient hybrid segmentation method that builds upon the strengths of CNNs and transformers. The proposed method maintains the basic architecture of U-net (Ronneberger et al., 2015), but also embeds a parallel global attention scheme implemented using a sequence of multi-head self-attention (MHSA) blocks (Vaswani et al., 2017). The MHSA based parallel attention blocks constitute the *transformer encoding path* in addition to the *CNN-encoding path* of U-net. Further, and more importantly, the transformer- and the CNN- encoding paths are connected via miniaturized bidirectional bridges that enable exchange of global and local image features between the two encoding paths. These bidirectional bridges are implemented as cross-attention blocks based on (Chen et al., 2021b), where these bidirectional bridges were used to obtain a hybrid transformer based MobileNet architecture for classification (Howard et al., 2017). The transformer encoding path acting in parallel to the CNN-encoding path, and the cross-attention based bidirectional bridges result in the proposed Hybrid Ladder Transformer Network (HyLT). We experiment on two publicly available datasets to study the segmentation performance of our network and show that HyLT achieves competitive results compared to pure CNN-, pure transformer- and hybrid architectures.

## 2. Methods

In this section, we describe the various components of the proposed hybrid ladder transformer for medical image segmentation, which is illustrated at a high level in Figure 1.

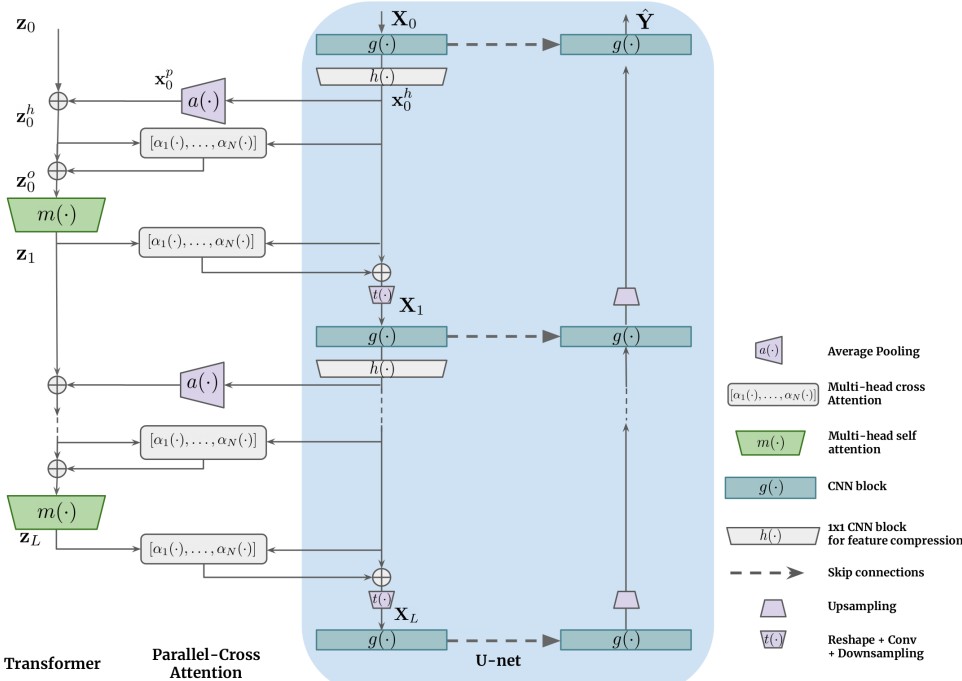

Figure 1: High level overview of the proposed Hybrid Ladder Transformer (HyLT). The grey inset box highlights a U-net like encoder-decoder architecture based on CNNs with skip connections of depth $L$. The CNN blocks in the encoder path consist of convolution layers with non-linearities that are able to extract local image features. The transformer part of the network operates in parallel to the CNN encoder path with randomly initialized tokens with additional patch level image features (obtained using the average pooling operation) as input in order to capture global image features. The tokens are injected with local image features via a cross-attention module that connects the CNN encoder path at each layer. The global features learned by the transformers are injected back into the local image features in the CNN encoder path via the cross-atention module in the second step in the ladder set-up. At each layer $l$, two cross attention modules operate as bridges between the CNN- and the transformer- blocks resulting in the *ladder* architecture.

The CNN block at layer $l$, $g_l(\cdot)$, of the CNN-encoding path takes an image, $\mathbf{X}_l \in \mathbb{R}^{H_l \times W_l \times F_{l-1}}$, of height $H_l$, width $W_l$ with $F_{l-1}$ features as input, and outputs an intermediate image array with increased features, $\mathbf{X}_l^h \in \mathbb{R}^{H_l \times W_l \times F_l}$, given in Eq. (1). These features are also used along the skip connections to increase the features at the decoder, similar to the standard U-net architecture. At the CNN-encoding path, these increased features are compressed to match the size of transformer token embedding size $E$ with 1x1 convolution block and spatially flattened into a vector, $\mathbf{x}_l^h \in \mathbb{R}^{H_l \cdot W_l \times E}$, given by $h_l(\cdot)$ in Eq. (2). These flattened image vectors are further split into $M$ patches and summarised using average patch level representations using an average pooling operation, $a_l(\cdot)$, resulting in $\mathbf{x}_l^p \in \mathbb{R}^{M \times E}$ in Eq. (3).

The transformer encoding path at level $l$ consists of updating $M$ tokens of size $E$, $\mathbf{z}_l \in \mathbb{R}^{M \times E}$. These tokens are first injected with image features $\mathbf{x}_l^p$ according to Eq. (4) to obtain intermediate tokens, $\mathbf{z}_l^h \in \mathbb{R}^{M \times E}$. The input tokens are then projected into the image

Table 1: System of equations used in the proposed Hybrid Ladder Transformer

$$\textbf{CNN block: } \mathbf{X}_l^h = g_l(\mathbf{X}_l), \in \mathbb{R}^{H_l \times W_l \times F_l} \tag{1}$$

$$\textbf{Feature compression: } \mathbf{x}_l^h = h_l(\mathbf{X}_l^h), \in \mathbb{R}^{H_l \cdot W_l \times E} \tag{2}$$

$$\textbf{Average pooling: } \mathbf{x}_l^p = a_l(\mathbf{x}_l^h), \in \mathbb{R}^{M \times E} \tag{3}$$

$$\textbf{Token update: } \mathbf{z}_l^h = \mathbf{z}_l + \mathbf{x}_l^p, \in \mathbb{R}^{M \times E} \tag{4}$$

$$\textbf{Local-to-global att.: } \mathbf{z}_l^o = \mathbf{z}_l^h + \left( [\alpha(\mathbf{z}_l \mathbf{Q}_l^n, \mathbf{x}_l^h, \mathbf{x}_l^h)]_{n=1:N} \right) \mathbf{M}_l^z, \in \mathbb{R}^{M \times E} \tag{5}$$

$$\textbf{Transformer block: } \mathbf{z}_{l+1} = m(\mathbf{z}_l^o), \in \mathbb{R}^{M \times E} \tag{6}$$

$$\textbf{Global-to-local att.: } \mathbf{x}_{l+1} = \mathbf{x}_l^h + \left( [\alpha(\mathbf{x}_l^h, \mathbf{z}_{l+1}^h \mathbf{K}_l^n, \mathbf{z}_{l+1}^h \mathbf{V}_l^n]_{n=1:N} \right) \mathbf{M}_l^x, \in \mathbb{R}^{H_l \cdot W_l \times E} \tag{7}$$

$$\textbf{Downsampling: } \mathbf{X}_{l+1} = t_l(\mathbf{x}_{l+1}) \in \mathbb{R}^{H_{l+1} \cdot W_{l+1} \times F_l} \tag{8}$$

$$\textbf{Attention: } \alpha(\mathbf{q}, \mathbf{k}, \mathbf{v}) = \sigma\left(\frac{\mathbf{q}\mathbf{k}^T}{\sqrt{E}}\right) \mathbf{v}, \text{ where } \sigma(\cdot) \text{ is the softmax operation} \tag{9}$$

feature space using the query projection matrix $Q \in \mathbb{R}^{E \times E}$ and combined with the image features using the $N$-head cross-attention operation in Eq. (5) forming a bridge from the CNN-encoding path to the transformer encoding path; this operation provides local image features to the global features learned by the transformer path. The multiple attention heads are linearly combined with the matrix, $\mathbf{M}_l^z \in \mathbb{R}^{N \cdot E \times E}$. After this cross attention operation, the tokens are further updated, $\mathbf{z}_{l+1} \in \mathbb{R}^{M \times E}$, using multi-head self attention (MHSA) operation (Vaswani et al., 2017), $m_l(\cdot)$, in Eq. (6).

The updated tokens from the transformer path, $\mathbf{z}_{l+1}$, are combined with image features, $\mathbf{x}_l$, forming a bridge from the transformer- to the CNN- encoding paths in the next step, given in Eq. (7). This is achieved by first projecting the tokens into the image feature space with the key-, value- projection matrices, $\mathbf{K}_l, \mathbf{V}_l \in \mathbb{R}^{E \times E}$, respectively. This is done $N$ times corresponding to multiple attention heads which are linearly combined with the matrix, $\mathbf{M}_l^x \in \mathbb{R}^{N \cdot E \times E}$. These updated image features with global information from the transformer-encoding path are reshaped back into the 2-d image space. Finally, another convolution operation and a downsampling operation is performed given as $t_l(\cdot)$ in Eq. (8), resulting in $\mathbf{X}_{l+1} \in \mathbb{R}^{H_{l+1} \times W_{l+1} \times F_l}$ as input to the next layer of the Hybrid Ladder Transformer.

The decoder is a standard U-net type decoder which receives features that have fused local and global features at each scale from the encoding path along the skip connections, and consists of convolution blocks and an upsampling operation. At the final layer of the decoder, the predicted segmentation masks are obtained, $\hat{\mathbf{Y}} \in \mathbb{R}^{H \times W}$. These predicted masks are compared with the corresponding binary ground truth segmentation, $\mathbf{Y} \in \mathbb{R}^{H \times W}$ during training and an appropriate loss is computed to optimize the weights of the Hybrid Ladder Transformer. In this work, we use Lovász-softmax loss which provides a tractable surrogate to optimize the intersection-over-union (IoU) in segmentation tasks (Berman et al., 2018).

Note that at input, for $l = 0$, the input image array $\mathbf{X}_0$ is the input image and the initial tokens $\mathbf{z}_0$ is a randomly initialized vector. The computations involved in computing the attention between vectors, $\mathbf{q}, \mathbf{k}, \mathbf{v}$ according to (Vaswani et al., 2017) is given in Eq. (9).

## 3. Data & Experiments

### 3.1. Data

The segmentation performance of the proposed model was evaluated with experiments on two publicly available histology datasets.

**GlaS** (Sirinukunwattana et al., 2017): This dataset comprises microscopy images of H&E stained slides for detection of malignant tumours in glands, known as adenocarcinomas which are some of the most prevalent form of cancer. The dataset consists of 165 images of resolution 775x552px of spatial resolution with $0.625\mu$m pixel resolution. The dataset was divided into training-, validation- and test- sets with 72, 13, 80 images, respectively.

**MoNuSeg** (Kumar et al., 2017): The H&E stained tissue images acquired with 40x microscope from several patients with tumours from The Cancer Genome Atlas (TGCA) program are used to create this dataset. Nuclei annotations from multiple organs and patients are provided as part of the dataset. The dataset consists of 30 images for training/validation and 14 for testing purposes. Each image is of 1000x1000px resolution and consists of several thousand nuclei annotations per image.

Four training examples and the corresponding binary masks for both the GlaS- and MoNuSeg- datasets are shown in Figures 3 and 4 in Appendix A.

### 3.2. Experimental set-up

We compare the performance of our method with relevant CNN- and transformer- based methods (Dosovitskiy et al., 2021). Specifically, we compare to the following methods: U-net (Ronneberger et al., 2015), hybrid dense prediction transformers (DPT Hybrid) (Ranftl et al., 2021), Transunet (Chen et al., 2021a) and Swin-UNet (Cao et al., 2021).

All methods were trained to minimize the Lovász-softmax loss (Berman et al., 2018). Performance of the different methods were compared using Dice overlap coefficient and mean IoU (MIoU) over the test set. All the models were implemented in PyTorch (Paszke et al., 2019), trained with stochastic gradient descent (SGD) optimizer for a maximum of 300 epochs with an initial learning rate of $10^{-3}$. Further details of the training hyperparameters are reported in Table 7 in Appendix B. The hyperparameters of the proposed model that describe additional details such as the number of filters per layer are reported in Table 6 in Appendix B.

### 3.3. Results

**Segmentation performance:** Test set segmentation performance on the two datasets for all methods is reported in Table 2. Our method, reported as **HyLT**, shows large improvements in both Dice overlap and MIoU scores across both datasets and compared to all methods. HyLT achieves a Dice score of 90.9±6.7% on GlaS- and 80.2±2.8% on MoNuSeg-datasets, respectively. Similarly, our method outperforms the baseline methods in MIoU with 83.9±10.16 and 67.1±3.9 on the GlaS- and MoNuSeg datasets. Qualitative results on two test images from both datasets, along with predictions from all methods are shown in Figure 2. The predictions from the models are overlaid with the input image and pixels are colour coded to indicate false negatives (green), false positive (red) and true positive (yellow). HyLT is able to obtain more complete segmentations in the larger structures present

Table 2: Test set performance of the different methods compared to the proposed Hybrid Ladder Transformer (HyLT) on the GlaS- and MoNuSeg- datasets. Dice overlap and mean intersection-over-union (MIoU) metrics are reported.

| Model | Parameters | Glas | | MoNuSeg | |
|---|---|---|---|---|---|
| | | Dice (%) | MIoU | Dice (%) | MIoU |
| U-net (Ronneberger et al., 2015) | **34.52M** | $84.45 \pm 8.40$ | $73.95 \pm 11.93$ | $79.30 \pm 3.13$ | $65.81 \pm 4.35$ |
| Hybrid DPT (Ranftl et al., 2021) | 124.00M | $81.80 \pm 11.37$ | $70.64 \pm 14.94$ | $66.31 \pm 4.97$ | $46.62 \pm 4.33$ |
| TransUNet (Chen et al., 2021a) | 143.62M | $78.98 \pm 13.22$ | $67.00 \pm 16.05$ | $69.63 \pm 2.89$ | $53.49 \pm 3.44$ |
| Swin-UNet (Cao et al., 2021) | 41.38M | $72.00 \pm 9.45$ | $57.05 \pm 10.80$ | $73.54 \pm 6.53$ | $58.43 \pm 5.44$ |
| HyLT (No transformer encoder) | 41.65M | $87.69 \pm 8.18$ | $78.95 \pm 12.02$ | $79.96 \pm 3.68$ | $66.77 \pm 5.08$ |
| **HyLT(Ours)** | 43.25M | $\mathbf{90.86 \pm 6.61}$ | $\mathbf{83.9 \pm 10.16}$ | $\mathbf{80.25 \pm 2.80}$ | $\mathbf{67.11 \pm 3.96}$ |

Table 3: Effect of varying the number of tokens (M), or equivalently the extent of patching of image features used as input to the transformer part of the proposed method. The patching is performed using the average pooling operation in Eq. (3) and the number of rows and columns are reported alongside the different values of $M$

| $M$ (row x col) | 2 (1x2) | 4( 2x2) | 6 (2x3) | 9 (3x3) | 16 (4x4) | 25( 5x5) | 36 (6x6) | 49 (7x7) |
|---|---|---|---|---|---|---|---|---|
| IoU | 81.1±12.5 | 82.7±11.9 | 81.2±13.5 | 83.7±10.1 | 83.5±11.0 | **83.9±10.2** | 83.7±10.5 | 83.5±11.0 |
| Dice | 89.0±8.3 | 90.0±8.1 | 83.7±10.1 | 90.8±6.4 | 90.6±7.7 | **90.9±06.6** | 90.7±6.8 | 90.6±7.2 |

in GlaS dataset compared to other methods with very few false negative pixels. In the MoNuSeg dataset, HyLT is again able to segment all the structures with few false positive predictions.

**Influence of number of tokens:** One important hyperparameter for our model is the number of tokens, $M$, used in the transformer blocks. As these tokens also obtain patch level image features as input, the number of patches provided from the average pooling operation in Eq. (3) to the transformer part of the model is also the same as $M$. This hyperparameter was tuned based on experiments performed on the GlaS dataset for different configurations. Test set results for different values of $M$ for the GlaS dataset are reported in Table 3. The best validation performance was obtained for $M = 25$ composed of 5x5 patches, and was also used for the MoNuSeg dataset.

## 4. Discussion & Conclusions

**Local- & global- features fusion**: The segmentation performance reported in Table 2 shows that the proposed Hybrid Ladder Transformer (HyLT) performs better than all comparing methods, in both Dice overlap and MIoU metrics. HyLT shows an improvement compared to U-net (Ronneberger et al., 2015) (pure CNN model) and Swin-Unet (Cao et al., 2021) (pure transformer model). This could be because pure CNN and pure transformer models have complementary capabilities – while CNNs focus on local features, transformers could attend to more global features (Raghu et al., 2021). This understanding has been further investigated in recent literature resulting in some hybrid models that combine CNNs and transformers in various ways, attempting to capture both local and global features. We compare to two of these hybrid models: hybrid DPT (Ranftl et al., 2021) and TransUnet (Chen et al., 2021a). HyLT also shows large improvements compared to these two hybrid models. While HyLT is also a hybrid model, the parallel transformer based encoding

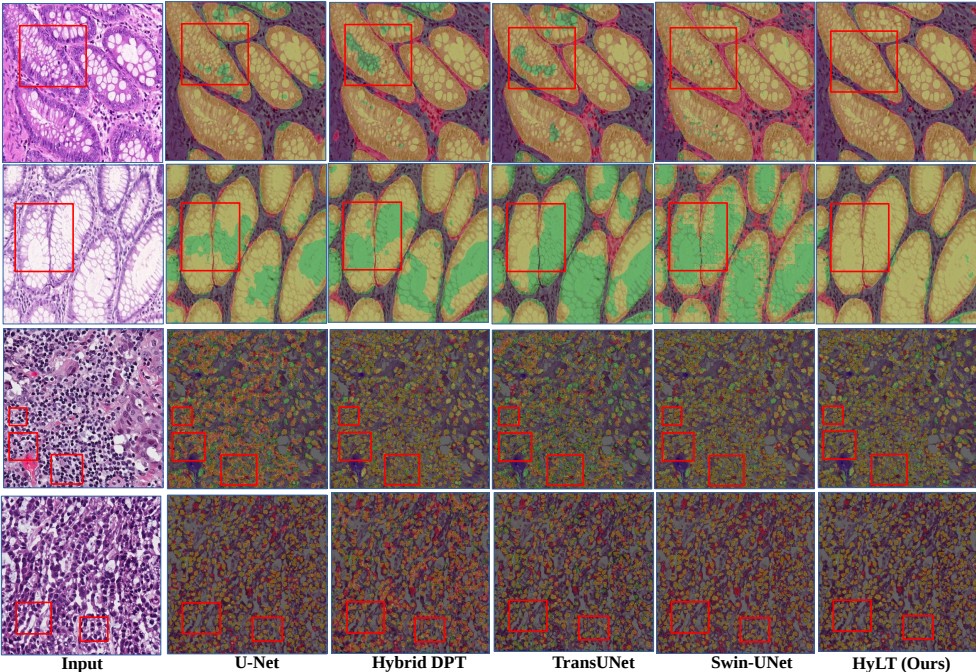

| Input | U-Net | Hybrid DPT | TransUNet | Swin-UNet | HyLT (Ours) |

Figure 2: Qualitative results on two sample test images from Glas- and MoNuSeg- datasets. The results are plotted in red for predicted regions that are false positive, green for false negatives, and true positive in yellow.The red box highlights regions where our method performs better than the other methods.

path and the mechanism of fusing local and global features with bidirectional cross-attention bridges enables HyLT to learn the most relevant combination of local and global features necessary for the segmentation task. This is different from the other two hybrid models which primarily use CNNs as shallow feature extractors and then extract deep features with the transformer parts of their model. The strengths of HyLT is highlighted in Figure 2 where it is able to handle both larger structures occurring in GlaS dataset, and smaller structures in MoNuSeg dataset with the same architecture and hyperparameters. This is in particular pronounced for the GlaS dataset where the structures of interest come in varying sizes and shapes (Appendix A, Figure 3). Access to more global context could be more beneficial for the GlaS dataset and the results in the first two rows in Figure 2 confirm this hypothesis, where we notice that HyLT is able predict more complete segmentation masks without an increase in false positives compared to other methods.

To further highlight the importance of the global feature extraction of the transformer encoding path, we report an ablated model without the transformer encoding path in HyLT and notice a considerable reduction in performance on GlaS dataset. There is no large reduction in performance on MoNuSeg dataset, which is to be expected as the structures in this dataset are small and primarily the decisions are driven by local features.

**Robustness to noise:** We performed additional experiments to test the robustness of segmentation to noise in the input images. This is in order to test the hypothesis that transformers are able to capture more global image features that are resistant to local noise (Raghu et al., 2021). We first train the images on clean images and at test time add

Table 4: Effect of noise on the segmentation performance for different extents of random masking (RM) of the image data. The performance degradation measure in Eq.(10) is reported (lower is better).

| Model | 10% RM | 20% RM | 30% RM |
|---|---|---|---|
| UNet (Ronneberger et al., 2015) | 17.62±13.23 | 22.27±13.72 | 25.27±13.54 |
| TransUNet (Chen et al., 2021a) | 13.31±9.56 | 20.81±11.67 | 25.47±11.87 |
| Hybrid DPT (Ranftl et al., 2021) | 11.66±7.00 | 19.29±8.98 | 24.18±9.94 |
| SwinUNet(Cao et al., 2021) | 8.97± 5.29 | 17.70±8.04 | 24.82±9.21 |
| **HyLT (ours)** | **8.56±7.16** | **16.40±9.28** | **22.79±11.13** |

noise to the test set images to evaluate the robustness to noise. We capture the robustness using a relative performance measure that captures the extent of performance degradation, $P_d$, given by

$$P_d(\mathbf{Y}_c, \mathbf{Y}_n, \mathbf{Y}) = \frac{|f(\mathbf{Y}_c, \mathbf{Y}) - f(\mathbf{Y}_n, \mathbf{Y})|}{f(\mathbf{Y}_n, \mathbf{Y})} \qquad (10)$$

where $\mathbf{Y}, \mathbf{Y}_c, \mathbf{Y}_n$ are the ground truth, predictions for clean images, predictions for noisy images, respectively. $f(\cdot)$ can be any performance metric; in our reporting we use Dice coefficient. In cases without any performance degradation, $P_g = 0$. Results for these experiments are reported in Table 4 for different extents of random masking (RM) where large regions of the image data are masked off. For the case of random mask noise of different extents (10%,20%,30%), we observe that our model has the least performance degradation compared to other methods. The performance degradation of Swin-UNet (Cao et al., 2021), which is a pure transformer based hierarchical model, is the closest to HyLT and could point to the strength of performing hierarchical attention at different scales that is used in both methods.

**Lightweight hybrid model**: The large model size of many vision transformers is because of the number, and embedding size, of the tokens used per image patch. We alleviate this by using a fixed number of global tokens ($M$) following (Chen et al., 2021b) to make the transformer encoding path lightweight. Further, the embedding size of these global tokens are fixed ($E$); when combining the tokens with the CNN features from the CNN-encoding path, we compress the CNN feature maps to match the embedding size of the global tokens (Eq. (8)). These strategies keep the complexity of HyLT (43.25M) lower compared to other hybrid models such as hybrid DPT (124M) and TransUnet (143M) as seen in Table 2. In fact, the transformer-encoding path uses only about 5% of the total parameters as shown with the ablated model (41.65M).

In conclusion, we presented the hybrid ladder transformer for biomedical image segmentation. This method aimed to combine the strengths of a hierarchical encoder-decoder based CNN model such as the U-net with transformers to obtain a hybrid segmentation model that is able to fuse local and global features. We used bidirectional cross-attention bridges at multiple resolutions for exchange of local and global features between the CNN- and transformer- encoding paths. Our experiments on two diverse datasets have demonstrated this strategy to be useful. The fusion of local and global features also render HyLT robust compared to other CNN-, transformer- and hybrid- methods to image perturbations. By fixing the size of the global tokens we also make our model efficient compared to other hybrid transformer methods.

## Acknowledgments

The results showed here are in part based upon data generated by the TCGA Research Network: https://www.cancer.gov/tcga. The authors would like to thank the Medical Image Analysis group at UCPH for fruitful discussions that have helped shape this manuscript.

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

## Appendix A. Additional visualizations

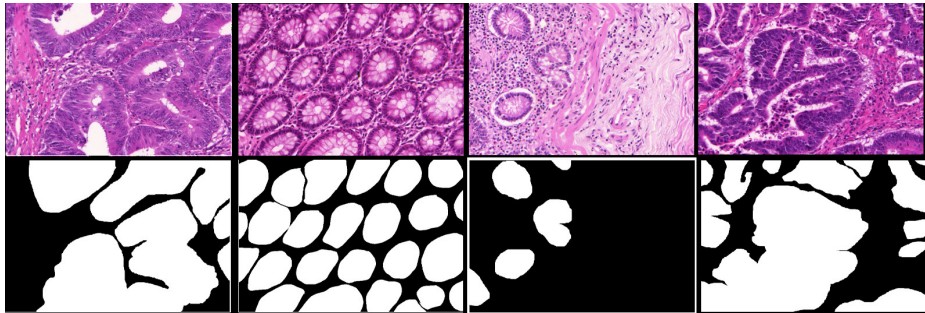

Figure 3: Four training cases from the GlaS dataset(Sirinukunwattana et al., 2017) along with the corresponding ground truth segmentations (bottom row), showing the diversity of the sizes of structures of interest in the stained histology images.

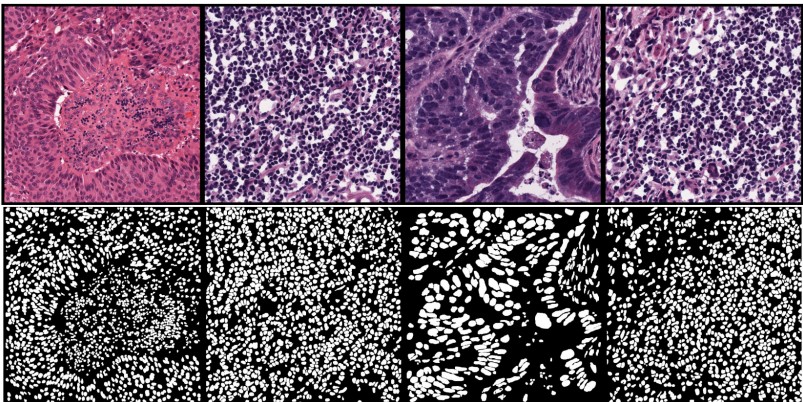

Figure 4: Four training cases from the MoNuSeg dataset(Kumar et al., 2017) along with the corresponding ground truth segmentations (bottom row). As with the GlaS dataset the structures of interest come in different sizes.

## Appendix B. Model hyperparameters

Table 5: Architecture of the hybrid transformer encoder in the proposed method

| Stage | Input Size | Operator |
|-------|-----------|----------|
| **Block1** | $224^2$x3 | Block1/Convblock |
| | $224^2$x64 | Block1/Compression |
| | $224^2$x64 | Block1/DoubleBridge |
| | $224^2$x64 | Block1/Outconv&Maxpooling |
| **Block2** | $112^2$x64 | Block2/Convblock |
| | $112^2$x128 | Block2/Compression |
| | $112^2$x128 | Block2/DoubleBridge |
| | $112^2$x128 | Block2/Outconv&Maxpooling |
| **Block3** | $56^2$x128 | Block3/Convblock |
| | $56^2$x256 | Block3/Compression |
| | $56^2$x192 | Block3/DoubleBridge |
| | $56^2$x256 | Block3/Outconv&Maxpooling |
| **Block4** | $28^2$x256 | Block4/Convblock |
| | $28^2$x512 | Block4/Compression |
| | $28^2$x192 | Block4/DoubleBridge |
| | $28^2$x512 | Block4/Outconv&Maxpooling |
| **Block5** | $14^2$x512 | Block5/Convblock |
| | $14^2$x1024 | Block5/Compression |
| | $14^2$x192 | Block5/DoubleBridge |
| | $14^2$x1024 | Block5/Outconv&Maxpooling |

Table 6: Additional model hyperparameters

| Parameter | Value |
|-----------|-------|
| # of layers ($L$) | 5 |
| # of tokens ($M$) | 6 |
| Token embedding size ($E$) | 192 |
| # of attention heads ($N$) | 6 |

Table 7: Training hyperparameters

| Optimizer | Lr | Lr_scheduler | Lr_decay_milestones | Lr_decay_gamma |
|-----------|-----|--------------|---------------------|----------------|
| SGD | 0.001 | MultiStepLR | [20, 50] | 0.9 |
| **Epoch** | **Input size** | **Batch_size** | **Crop_size** | **Color_jitter_params** |
| 300 | 224*224 | 8 | [32,32] | [0.1, 0.1, 0.1, 0.1] |

