# OpenReview forum: "Hybrid Ladder Transformers with Efficient Parallel-Cross Attention for Medical Image Segmentation"
_MIDL.io/2022/Conference — MIDL 2022_

### Official Review · Reviewer_Pw3i · 2022-01-22

**Confidence:** 4
**Preliminary Rating:** 3
**Recommendation:** Poster

**Summary:**

The paper proposes a hybrid ladder transformer (HyLT), which is a hybrid transformer model that works in conjunction with a CNN. Specifically,  HyLT uses learnable global attention heads along with the traditional UNet to encode long-range dependencies. Experiments are conducted across two datasets where HyLT achieves better performance than standard baselines.

**Strengths:**

1) The motivation of using both CNNs and Transformers to encode both local and long range features is a good direction, especially for medical image segmentation.

2) The experiments conducted and the ablation study conducted to validate the performance of HyLT is good. So much analysis has been conducted and reported making the paper strong in terms of validation.

3) The reported segmentation accuracy is a significant improvement over the baselines.

**Weaknesses:**

1) One thing that is not clear about the method is that how actually can the authors say both local and global features are extracted? Because the architecture has cross-attention block which acts between convolutional features and transformer features and these are injected into the next layer to both the conv-block as well as MHSA. With local features being fed to MHSA and global features being fed to conv-blocks, isn't the intuition kind of against the motivation?

2)  From the experiments and results, it looks like UNet is the second-best performing method out of all the recent baselines. I find this a bit glaring because there have been many conv networks like UNet++, DenseUNet which easily outperform UNet. Also, almost all the recent transformer-based methods like TransUNet, MedT, UTNet have also shown better performance than UNet. So, I find it difficult to understand why the number of TransUNet and SwinUNet are below that of the UNet. Have the authors trained the baselines well? Note: As the datasets used in the experiments are very small, the authors might need to reduce LR from the original values while training the transformer baselines.



**Deanonymize Review:**

no

**Final Rating After The Rebuttal:**

3: Borderline

**Justification Of The Final Rating:**

I think the authors have responded well to reason transformer baselines do not perform well. However, I do not think the answer how both local- and global- features are extracted is valid. So, I keep my score unchanged.

**Paper Type:**

methodological development

**Questions To Address In The Rebuttal:**

1) The explanation of why the authors think that the proposed network extracts both local and global features and how it avoids the overlap training of CNN and Transformer.

2) Did the authors train the baselines properly and more details on the same.

**Special Issue:**

no

---

### Official Review · Reviewer_fo44 · 2022-01-24

**Confidence:** 3
**Preliminary Rating:** 4
**Recommendation:** Poster

**Summary:**

The paper presents results for two datasets of H&E stained slides, where the binary segmentation is better than any compared methods. The mechanism has its merit in fusing local and global features with bidirectional cross-attention parts, to handle both larger and smaller structures. It achieves this by combining CNN encodings of the image information at multiple levels via cross-attention blocks feeding from and into the encoder. The global information is learned by updating of the tokens via Multi Head Self Attention. Pooling is used to downsample the CNN features.





**Strengths:**

- The authors motivate the use by lying out the issues of the popular networks for segmentation and recapitulate the upcoming of the transformer for the task.
- Some relevant current related work is mentioned.
- Figure 1 gives a helpful overview of the model.
- The architecture is extensively described and hyperparameters are reported.
- Influence of number of tokens and input noise is reported.


**Weaknesses:**

- “The increase in the model complexity (number of parameters)” of methods in the introduction are claimed to be addressed in the HyLT but the number of parameters are 9.53M for the UTNet and for MedT only 1.4M, according to their papers while a number of 41.65M  is reported here by the authors
- The results table reports “HyLT (No transformer encoder)” -  why does it have such good performance compared to the other methods when it is just some CNN? That does not seem in line with the argumentation of the benefit of HyLT
- “HyLT is able to obtain more complete segmentations in the larger structures present in GlaS dataset compared to other methods with very few false negative pixels.” but images show partly rather small parts of the segmentation of the other models missing?
- Why was the comparison made performed with Hybrid DPT and not instead also with UTNet, even though it is introduced as suitable in the introduction? Furthermore, a comparison to the 2D nnUNet should be included, which has been established as the most competitive baseline in the field.
And it seems worth trying out to just increase receptive field or incorporate information about neighboring cells in a simpler way for the nuclei segmentation task since there is no need for the context of larger image parts?


**Deanonymize Review:**

no

**Detailed Comments:**

After the summary of CNNs in the abstract (this is more suitable for the introduction and could be shortened so the actual work of the paper is the focus of the abstract, so it becomes clear what is done differently to previous application of the common transformer networks) the main sentence  “approach for medical image of segmentation tasks and we propose a hybrid transformer model that works in conjunction with a CNN” has a mix up, it should probably read “for segmentation of medical images”.
Check the references, some cited as preprint are to be updated
Perhaps the inset coloring of the Unet should not be the same as the MHCA blocks
It can be confusing, that the description says “back into the local image
features in the CNN encoder path via the second cross-attention module” however it is the fourth MH cross-attention block that feeds back into the encoder. It might be the second “step” of the “ladder” but not the second block as was defined in the authors' terminology
The dimension of the output of the 1x1 convolution is noted in the description as Hl·Wl×F  but should be x E instead of the same dimension as the input of it, since it is supposed to match the Embedding dimension
Similarly, the description says “The intermediate tokens are projected into the image feature space using the query projection matrix Q” while formula (5) multiplies zl with it even though the intermediate tokens are “z_l^h” in the text. Also image features in the text are “xp^l” but not in the formula.
Spelling “Dice socre”
Putting the image description in the Results section text might hinder reading flow (“The predictions from the models are overlaid with the input image and pixels are colour coded to indicate false negatives (green), false positive (red) and true positive (yellow).”)
The results images could be better visible in terms of coloring



**Final Rating After The Rebuttal:**

4: Weak Accept

**Justification Of The Final Rating:**

I appreciate the responses of the authors, in particular pointing out the ablation study of HyLT without transformer and adding the more competitive nnUnet baseline. I do not fully share the view that this comparison is not "fully fair". The nnUNet framework usually only applies very straightforward rules about number of encoder-decoder levels, pre-processing and feature channels. The fact that it nearly matches the results of HyLT demonstrates once more that even very thoughtful new architectures cannot squeeze out substantially better performance (I'll admit that the training pipeline of the nnUNet is quite elaborate). I stick with my "weak accept" recommendation because I am confident that this paper can make a nice contribution to MIDL.

**Paper Type:**

methodological development

**Questions To Address In The Rebuttal:**

The authors say “constraints on fixed input image sizes in these methods are some of the limitations we aim to address in our work”. Maybe it can actually be extended to data that is not 2D in the future
What is the performance of the 2D nnUNet on the same data https://github.com/MIC-DKFZ/nnUNet

**Special Issue:**

no

---

### Meta-Review · Area_Chair_puQd · 2022-02-21

**Recommendation:** Accept (Poster)
**Confidence:** 5

**Metareview:**

The authors propose a way to combine Transformers and CNNs. Comparisons with nnUNet are essential as it wins many challenges.

---

### Decision · Program_Chairs · 2022-02-28

Accept